# Analysis of Vibration Characteristics of Planetary Gearbox with Broken Sun Gear Based on Phenomenological Model

**Mengting Zou [1,2], Jun Ma [1,2,*], Xin Xiong [1,2] and Rong Li [1,2]**

[1]   Faculty of Information Engineering and Automation, Kunming University of Science and Technology, Kunming 650500, China; m18503478169@163.com (M.Z.)

[2]   Yunnan International Joint Laboratory of Intelligent Control and Application of Advanced Equipment, Kunming University of Science and Technology, Kunming 650500, China

[*]   Correspondence: mjun@kust.edu.cn; Tel.: +86-15087151416

**Abstract:** To investigate the vibration properties in healthy and fault conditions of planetary gearboxes, a phenomenological model is constructed to present the vibration spectrum structure. First, the effects of the base deflection of the gear fillet and the flexibility between the root circle and the base circle on the time-varying meshing stiffness are considered in order to construct an equivalent model of time-varying mesh stiffness and broken tooth faults, exploring the law of variation for meshing stiffness when differently sized faults occur on the sun gear. Then, considering both the effect of the vibration transfer path and the meshing impacts, we establish phenomenological models of planetary gears under healthy and fault conditions. By comparing and analyzing the phenomenological model based on the cosine function to verify the effectiveness of the proposed model. The experimental results show that the error of the proposed model is 1.38% lower than that of the traditional phenomenological model, and the proposed model can accurately analyze the frequency, amplitude, and sideband characteristics of the vibration signals of sun gear with different degrees of broken tooth, which can be used for the local fault diagnosis of planetary gearboxes.

**Keywords:** gear fault; phenomenological model; vibration feature

## 1. Introduction

Planetary gearboxes (PGs) are widely used in aerospace, wind power generation, and other fields due to their small size, smooth transmission, large transmission ratio, and high efficiency. However, due to the unique characteristics of the transmission structure, the complexity of the vibration transmission path of PGs, and the harsh working environment, its gears, bearings, and other key components are prone to different types and degrees of failure, resulting in the degradation of gearbox performance and even causing major accidents. Therefore, the fault diagnosis of PGs is of great significance in avoiding safety accidents.

In recent years, the rapid development of numerical simulation technology has provided convenient and scientific solutions for clarifying the vibration spectrum structure and carrying out research on fault diagnosis methods of PGs [1]. Two different models are used to study the vibration features of PGs: the lumped-parameter dynamic model (LPDM) and the phenomenological model. Based on LPDM, nonlinear factors in the gearbox, such as shaft misalignment [2], manufacturing errors [3], backlash [4], lubrication [5], temperature [6] and friction force [7], have been investigated in depth. However, solving the equations of motion in LPDM is extremely time-consuming and complicated due to a large number of parameters and freedom degrees in this model. To reduce the parameters of the dynamics modeling process and computational complexity, scholars proposed to describe the vibration signal of the gearbox as a function related to the meshing frequency and then construct a simple and efficient phenomenological model of the gearbox [8].

Zhou et al. [9] proposed a representative phenomenological model to formulate and illustrate the relationship between fault-induced modulation patterns and characteristic spectral distributions. Yu et al. [10] established a phenomenological model considering the time-varying speed conditions of the planetary gearbox, which effectively explained the sideband symmetry near the resonance frequency. Liu et al. [11] established a phenomenological model of the planetary gearbox considering uneven load distribution among planetary gears and revealed the mechanism of additional sidebands under nonuniform load distribution conditions. Parra et al. [12] found the phenomenological model and the centralized parameter model for PGs without and with faults, respectively, and compared the results of these two models to confirm that the phenomenological model is the most efficient in generating gearbox vibration signals. However, the phenomenological model has some drawbacks. Because the phenomenological model is constructed based on trigonometric functions, there is no impact component in the synthesized vibration signal, which cannot reflect the meshing impact characteristics better. To overcome this drawback and further improve the performance and adaptability of the phenomenological model, Luo et al. [13,14] unified the reference points of the stiffness model and the phase model and proposed an improved phenomenological model based on the consideration of transient shock effects to realize the simultaneous characterization of the modulation characteristics and shock characteristics of the gearbox vibration signal and applied it to characterize the gearbox vibration mechanism when different sizes of faults occur on the sun gear. Liu et al. [15] calculated the impact force when the gears were meshed, established the phenomenological model considering the meshing impact, compared the effects of angular displacement and meshing influence on the vibration signal of the planetary gearbox, and confirmed that the angular displacement has a more significant effect on the gearbox spectrum. In addition, Zhang et al. [8] improved the traditional phenomenological model by considering the gear mesh amplitude's random magnitude and the transmission path's attenuation effect, revealing the spectrum's asymmetry. Xu et al. [16] used the projection of the triangular function to represent the time-varying direction of the impact force in the sensor measurement direction and the Fourier series to represent the transmission path in order to establish the phenomenological model of the local fault of the sun gear, effectively deriving the resonance modulation sidebands of PGs. Nie et al. [17] developed a phenomenological model of a healthy wind turbine gearbox and used the Hanning function to represent the effect of time-varying transfer paths to reveal the vibration frequency characteristics of a multistage gearbox system.

In summary, the research results on the modeling and analyzing of the phenomenological model provide good support for revealing the vibration mechanism and fault diagnosis of PGs. An in-depth analysis of the internal structure and operating mechanism of the planetary gearbox reveals that the accurate calculation of the time-varying mesh stiffness is the basis for the construction of the phenomenological model of the gearbox. In contrast, the existing phenomenological model ignores the influence of the fillet base deflection and inter-gear flexibility on the mesh stiffness when using the potential energy method to calculate the mesh stiffness, which increases the model characterization error. In addition, the mesh shock generated during gear meshing affects the vibration signal. However, the existing phenomenological model is weak in portraying the shock component of the vibration signal and cannot reflect the pulse modulation characteristics caused by the meshing shock. In this paper, based on the time-varying meshing stiffness, the meshing shock function and the influence of the vibration signal transmission path are considered in order to construct a more realistic phenomenological model for PGs, which provides a theoretical basis for the fault diagnosis of PGs.

## 2. Vibration Mechanism of Planetary Gearbox

### 2.1. Time-Varying Meshing Stiffness

Time-varying mesh stiffness is a periodic internal excitation of gear pairs, which is very important for the vibration analysis of gear systems [18]. Considering the base deflection of

the gear fillet and the influence of flexibility between the root circle and the base circle [19], the gear is simplified to a variable-section cantilever beam fixed on the gear base circle (as shown in Figure 1), and the modified potential energy method is used to calculate the time-varying meshing stiffness of the planetary gearbox for two cases, i.e., the gear root circle being smaller or bigger than the base circle, respectively [20]. According to the knowledge of principle of mechanics, the radius of gear base circle $R_b$ and root circle $R_f$ can be calculated as follows [21]:

$$R_b = \frac{mz}{2}\cos(\theta) \tag{1}$$

$$R_f = \frac{mz}{2} - (h^*_a + c^*)m \tag{2}$$

where $m$, $z$, and $\theta$ represent module, number of teeth, and pressure angle, respectively. $h^*_a$ and $c^*$ are addendum and tip clearance coefficients.

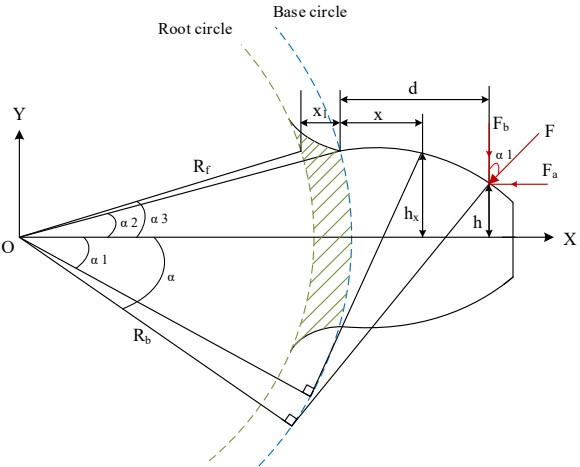

**Figure 1.** The cantilever beam model of gear teeth.

When the gear root circle is bigger than the base circle, the equations of $k_b$, $k_s$, $k_a$, and $k_h$ are expressed as follows [21]:

$$k_{b,normal} = \frac{1}{\int_{-\alpha_1}^{\alpha_4} \frac{3\{1+\cos\alpha_1[(\alpha_2-\alpha)\sin\alpha-\cos\alpha]\}^2(\alpha_2-a)\cos\alpha}{2EL[\sin\alpha+(\alpha_2-\alpha)\cos\alpha]^3}\mathrm{d}\alpha} \tag{3}$$

$$k_{s,normal} = \frac{1}{\int_{-\alpha_1}^{\alpha_4} \frac{1.2(1+v)(\alpha_2-\alpha)\cos\alpha\cos^2\alpha_1}{EL[\sin\alpha+(\alpha_2-\alpha)\cos\alpha]}\mathrm{d}\alpha} \tag{4}$$

$$k_{a,normal} = \frac{1}{\int_{-\alpha_1}^{\alpha_4} \frac{(\alpha_2-\alpha)\cos\alpha\sin^2\alpha_1}{2EL[\sin\alpha+(\alpha_2-\alpha)\cos\alpha]}\mathrm{d}\alpha} \tag{5}$$

$$k_{h,normal} = \frac{\pi EL}{4(1-v^2)}. \tag{6}$$

When the gear root circle is smaller than the base circle, the energy stored between the base circle and the root circle (the grid line portions as shown in Figure 1) needs to be increased on the original basis; then, Equations (3)–(5) are modified as follows [21]:

$$k_{b,normal} = \frac{1}{\int_{-\alpha_1}^{\alpha_2} \frac{3\{1+\cos\alpha_1[(\alpha_2-\alpha_1)\sin\alpha-\cos\alpha]\}^2(\alpha_2-a)\cos\alpha}{2EL[\sin\alpha+(\alpha_2-\alpha)\cos\alpha]^3}\mathrm{d}\alpha + \int_0^{r_b-r_f} \frac{[(d+x_1)\cos\alpha_1-h\sin\alpha_1]^2}{EI_{x_1}}\mathrm{d}x_1} \tag{7}$$

$$k_{s,normal} = \cfrac{1}{\int_{-\alpha_1}^{\alpha_2} \cfrac{1.2(1+v)(\alpha_2-\alpha)\cos\alpha\cos^2\alpha_1}{EL[\sin\alpha+(\alpha_2-\alpha)\cos\alpha]}d\alpha + \int_0^{r_b-r_f} \cfrac{(1.2\cos\alpha_1)^2}{GA_{x_1}}dx_1} \tag{8}$$

$$k_{a,normal} = \cfrac{1}{\int_{-\alpha_1}^{\alpha_2} \cfrac{(\alpha_2-\alpha)\cos\alpha\sin^2\alpha_1}{2EL[\sin\alpha+(\alpha_2-\alpha)\cos\alpha]}d\alpha + \int_0^{r_b-r_f} \cfrac{(\sin\alpha_1)^2}{EA_{x_1}}dx_1}. \tag{9}$$

The equation of $k_f$ can be expressed as follows [22]:

$$\frac{1}{k_f} = \frac{\cos^2\alpha_m}{EL}\left\{L^*\left(\frac{\mu_f}{S_f}\right)^2 + M^*\left(\frac{\mu_f}{S_f}\right) + P^*\left(1+Q^*\tan^2\alpha_m\right)\right\}. \tag{10}$$

where $\alpha_m$ is the pressure angle of contact point, $\mu_f$ and $S_f$ are defined as shown in Figure 2. The coefficients $L^*$, $M^*$, $P^*$, $Q^*$ can be approached by polynomial functions [23]:

$$X_i^* = A_i/\theta_f^2 + B_i h_{fi}^2 + C_i h_{fi}/\theta_f + E_i h_{fi} + F_i. \tag{11}$$

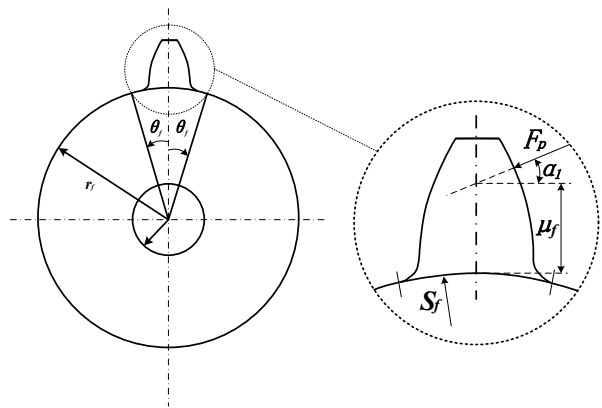

**Figure 2.** Foundation deformation calculation.

The values of $A_i$, $B_i$, $C_i$, $E_i$, $F_i$ are given in Table 1.

**Table 1.** Values of the coefficient of Equation (11).

|        | $A_i$ | $B_i$ | $C_i$ | $D_i$ | $E_i$ | $F_i$ |
|--------|-------|-------|-------|-------|-------|-------|
| $L^*$  | $-5.574 \times 10^{-5}$  | $-1.9986 \times 10^{-3}$ | $-2.3015 \times 10^{-4}$ | $4.7702 \times 10^{-3}$  | $0.0271$  | $6.8045$ |
| $M^*$  | $60.111 \times 10^{-5}$  | $-28.100 \times 10^{-3}$ | $-83.431 \times 10^{-4}$ | $-9.9256 \times 10^{-3}$ | $0.1624$  | $0.9086$ |
| $P^*$  | $-50.952 \times 10^{-5}$ | $185.50 \times 10^{-3}$  | $0.0538 \times 10^{-4}$  | $53.3 \times 10^{-3}$    | $0.2895$  | $0.9236$ |
| $Q^*$  | $-6.2042 \times 10^{-5}$ | $9.0889 \times 10^{-3}$  | $-4.0964 \times 10^{-4}$ | $7.8297 \times 10^{-3}$  | $-0.1472$ | $0.6904$ |

In summary, the total effective meshing stiffness of the planetary gearbox can be described as follows:

$$k_{total} = \begin{cases} 1/\left(\frac{1}{k_h} + \frac{1}{k_{b1}} + \frac{1}{k_{s1}} + \frac{1}{k_{a1}} + \frac{1}{k_{f1}} + \frac{1}{k_{b2}} + \frac{1}{k_{s2}} + \frac{1}{k_{a2}} + \frac{1}{k_{f2}}\right), \text{ single-tooth-pair} \\ \\ \sum_{i=1}^{2} \cfrac{1}{\frac{1}{k_{h,i}} + \frac{1}{k_{b1,i}} + \frac{1}{k_{s1,i}} + \frac{1}{k_{a1,i}} + \frac{1}{k_{f1}} + \frac{1}{k_{b2,i}} + \frac{1}{k_{s2,i}} + \frac{1}{k_{a2,i}} + \frac{1}{k_{f2}}}, \text{ double-tooth-pair} \end{cases} \tag{12}$$

In the expressions above, subscripts 1 and 2 indicate the driving and driven gears respectively, and *I* denotes the gear pair number in meshing.

Based on the above theoretical analysis, the trend of health gear meshing stiffness change under two cases is calculated as shown in Figure 3. In comparison, the time-varying mesh stiffness amplitude decreases by 18.75% when the gear root circle is smaller than the base circle. According to the literature [21], the mesh stiffness values obtained by

considering the base deflection of the gear fillet and the flexibility between the root circle and the base circle are more accurate compared with the mesh stiffness values calculated by the ISO standard, which lays the foundation for establishing a precise phenomenological model.

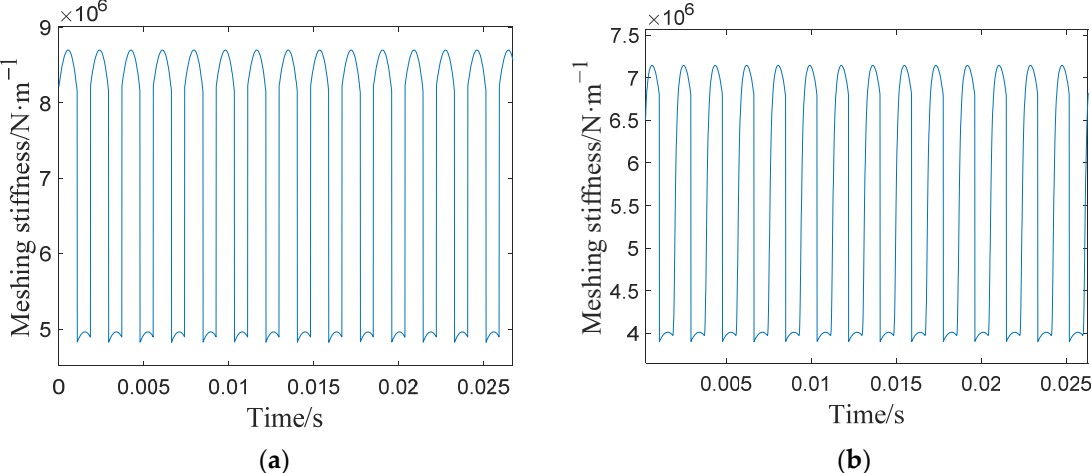

**Figure 3.** Meshing stiffness of the healthy planetary gear: (**a**) $R_f > R_b$; (**b**) $R_f < R_b$.

### 2.2. Meshing Stiffness of the Fault External Gear Pair

Based on the normal meshing stiffness variation, the meshing stiffness variation in the broken teeth of the sun gear is further investigated. Figure 4 is a simplified schematic diagram of the fractured tooth of the sun gear, $ds$ is the size of the broken tooth of the sun gear, the cross-section of the broken tooth intersects the tooth profile at point $D$, and point $K$ is the intersection of the tooth profile and the tooth apex circle. The meshing force in the case of a broken tooth is shown in Figure 5. In Figure 5, the gear pair enters meshing from point $A$. Point $C$ is an arbitrary meshing point on the tooth profile, and point $O$ is the rotation center of the sun gear. Theoretically, the meshing point of the external gear pair enters the mesh from point $A$ and exits from point $K$. However, due to the presence of the fault, the meshing process ends early at point $D$, resulting in the stiffness of the gear pair being 0 in the $DK$ range.

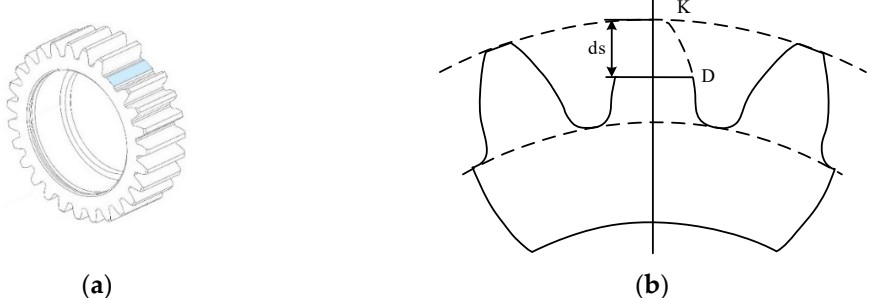

**Figure 4.** Schematic diagram of the sun gear with a broken tooth: (**a**) three-dimensional diagram; (**b**) two-dimensional diagram.

Based on the broken tooth mesh force distribution variation, the $k_b$, $k_s$, $k_a$, and $k_h$ are first calculated using Equations (13)–(16) and substituted into Equation (12) to calculate the mesh stiffness for different degrees of broken teeth and plot the variation curve (as shown in Figure 6).

$$k_{b,\,fault} = \begin{cases} k_{b,\,normal}, \alpha \leqslant \alpha_d \\ 0, \alpha > \alpha_d \end{cases} \tag{13}$$

$$k_{s,\,fault} = \begin{cases} k_{s,\,normal}, \alpha \leqslant \alpha_d \\ 0, \alpha > \alpha_d \end{cases} \tag{14}$$

$$k_{a,\,fault} = \begin{cases} k_{a,\,normal}, \alpha \leqslant \alpha_d \\ 0, \alpha > \alpha_d \end{cases} \tag{15}$$

$$k_{h,\,fault} = \begin{cases} k_{h,\,normal}, \alpha \leqslant \alpha_d \\ 0, \alpha > \alpha_d \end{cases} \tag{16}$$

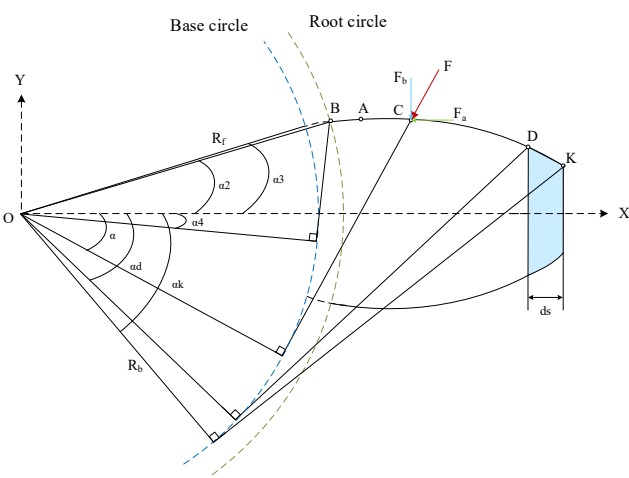

**Figure 5.** The meshing force on the broken sun gear tooth.

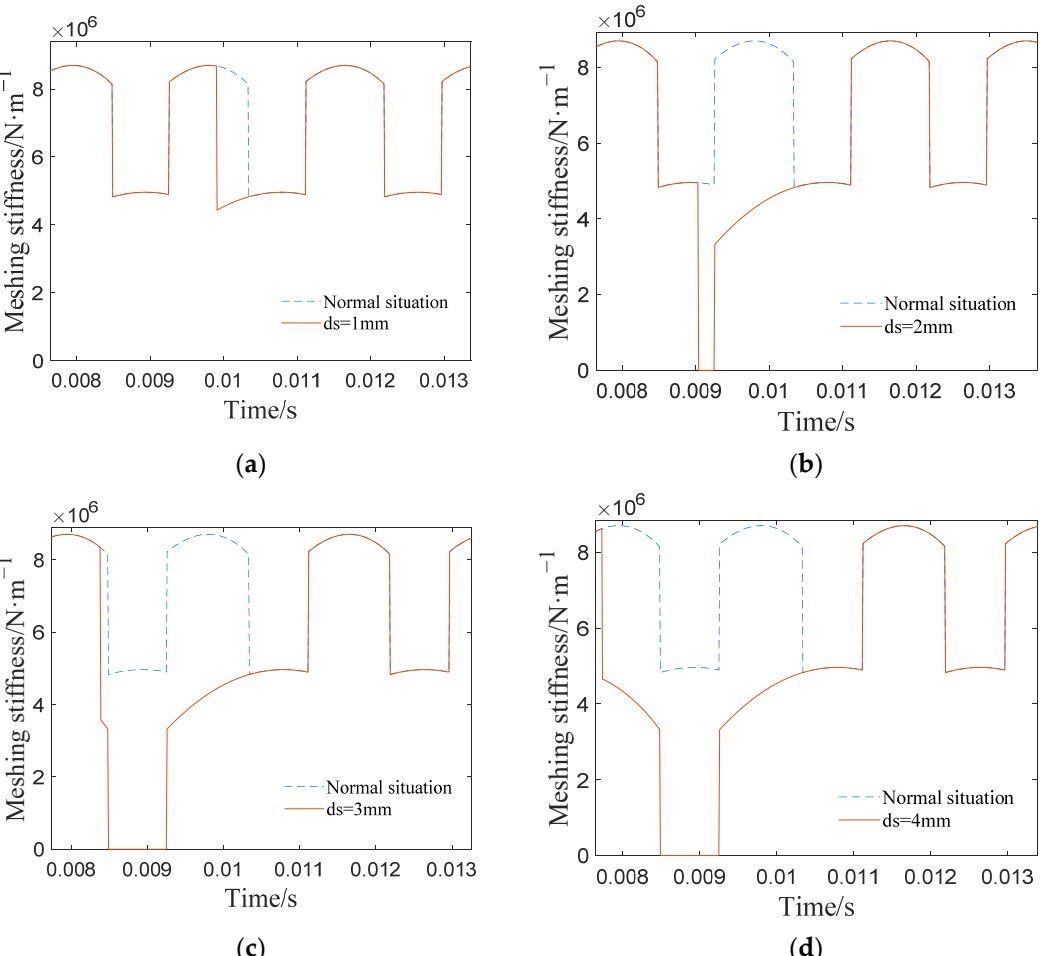

**Figure 6.** Stiffness curve of external gear pair under the broken tooth condition of sun gear: (**a**) ds = 1 mm(1/4); (**b**) ds = 2 mm(2/4); (**c**) ds = 3 mm(3/4); (**d**) ds = 4 mm(completely broken teeth).

The comparative analysis of Figure 6 shows that when the sun gear suffers a broken tooth fault, the meshing stiffness decreases in the range of broken teeth. With the increase in the fault size, the fall time of the gear pair is significantly earlier, and the meshing process is terminated earlier. It is also found that when the fault size is less than $\frac{1}{2}$ of the tooth height (ds $\leq$ 2 mm), the gear pair has a single tooth meshing zone. The variation law of this meshing stiffness with the fault degree provides theoretical support for constructing signal models characterizing different fault degrees.

## 3. Phenomenological Model of Gearbox with Broken Sun Gear Fault

### 3.1. Phenomenological Model of Planetary Gearbox under Healthy Condition

Gears generate one shock response per engagement at the instant of single versus double tooth meshing. Setting $\omega$ as the intrinsic frequency, the single-mesh shock response can be expressed as follows [24]:

$$r(t) = \frac{F}{\sqrt{k_{total}(t)}}\left(\frac{1}{m_{e1}} + \frac{1}{m_{e2}}\right)e^{-\xi\omega t}\sin(\omega t),\tag{17}$$

where $F$ donates the impulse, which is the integral of the meshing impact force in a single impact time, $k_{total}(t)$ is the time-varying meshing stiffness, $\xi$ is the damping coefficient, and $m_{e1}$ and $m_{e2}$ are the equivalent masses of the master and passive gears, respectively. In this paper, $F$ and $\xi$ are set as 0.139 N·s, and $-0.08$, $m_{e1}$ and $m_{e2}$ are designated as 0.2 kg and 0.5 kg, respectively. The intrinsic frequency is about 1400 Hz.

It is crucial to calculate the occurring moment of the meshing shock and its amplitude in order to establish an accurate phenomenological model with meshing impacts. However, since the planetary gearbox contains multiple gears, it is difficult to calculate these two values directly, which must be estimated indirectly through the meshing stiffness and phase model. So, based on the mesh stiffness calculation in Section 2, we use the initial meshing point as the reference point, and the vibration $v_{rpi}(t)$ and $v_{spi}(t)$ generated by the ring-planet and sun-planet meshing points can be expressed as follows [25]:

$$v_{rpi}(t) = \sum_{\substack{q=-Q \\ q \neq 0}}^{Q} V_q^{rp} e^{j2\pi q f_m t} e^{j(-2\pi q \gamma_{rs} + \varphi_q)}\tag{18}$$

$$v_{spi}(t) = \sum_{\substack{q=-Q \\ q \neq 0}}^{Q} V_q^{sp} e^{j2\pi q f_m t} e^{j\varphi_q}.\tag{19}$$

Because of the complexity of the vibration signals of the planetary gearbox, there are several possible transfer paths from the meshing point to the sensor, and the influence of the transmission paths on the vibration signal must be further considered. This paper mainly considers the six transmission paths, as shown in Figure 7 [26].

Path 1: sun-planet meshing point → sun gear shaft → gearbox housing → transducer;

Path 2: sun-planet meshing point → planet gear → ring gear → gearbox housing → transducer;

Path 3: sun-planet meshing point → planet gear → planet gear shaft → planet carrier → gearbox housing → transducer;

Path 4: ring-planet meshing point → sun gear → sun gear shaft → gearbox housing → transducer;

Path 5: ring-planet meshing point → planet gear shaft → planet carrier → gearbox housing → transducer;

Path 6: ring-planet meshing point → ring gear → gearbox housing → transducer.

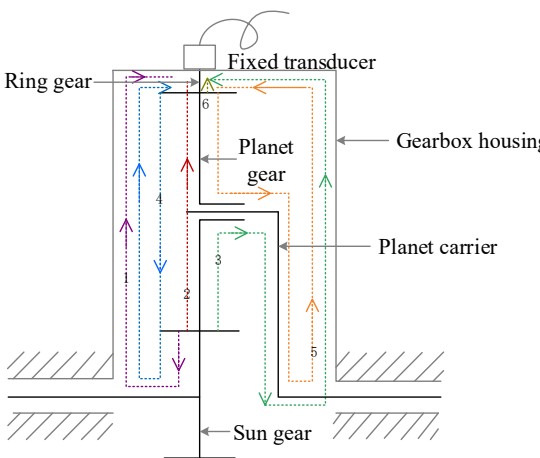

**Figure 7.** The potential transmission path of vibration signal of planetary gear.

From the description of the vibration transfer paths, it can be seen that the lengths of Path 1, Path 3, Path 4, and Path 5 are substantially unchanged, while the lengths of Path 2 and Path 6 vary with the carrier rotation. In other words, among the above six paths, Path 1, Path 3, Path 4, and Path5 are time-invariant, while Path 2 and Path 6 are time-varying. In Path 2 and Path 6, the transfer distance is periodically time-varying with the rotating carrier (as shown in Figure 8), and it then causes amplitude modulation on the response signal. The remaining four paths only affect the ratio of amplitude attenuation of the meshing vibrations and do not cause amplitude modulation influence on the response signal. Therefore, this paper mainly considers transfer Paths 2 and 6, and a Hanning function is used to consider the influence of the transfer path [27]. The modulation effects of transfer paths from the meshing point to the transducer is described as follows [28]:

$$W_{pi}(t) = \alpha \cdot [A - \cos(2\pi N f_c t)] / A. \tag{20}$$

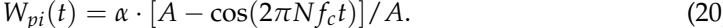

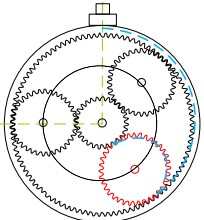 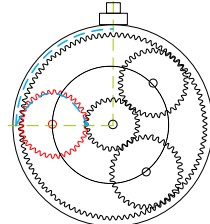 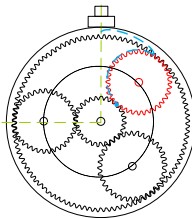

**Figure 8.** Propagating distance of different meshing positions.

Constant coefficient $\alpha$ is used to present the attenuation effect of the time-invariant part, $A$ is the amplitude coefficient of the transfer function, $N$ is the number of planets, and $f_c$ donates the rotation frequency of the carrier.

In summary, to simplify the computational complexity of the model, it is assumed that all gears are isotropic for vibration transmitting, and the vibration signal measured by the fixed transducer is a linear superposition of multiple internal and external meshing vibration signals, expressed as follows:

$$x(t) = \sum_{i=1}^{N} (v_{rpi}(t) + v_{spi}(t)) \cdot \Phi \cdot W_{pi}(t) \tag{21}$$

where $N$ is the number of planets, $v_{rpi}(t)$ and $v_{spi}(t)$ are the vibration signals of the planet-ring meshing and the sun-planet meshing, $\Phi$ is the meshing impact effect, and $W_{pi}(t)$ representing the modulation effect of the transmission path from the meshing point to the fixed sensor.

The flowchart of the overall numerical process for solving the proposed phenomenological model and obtaining the gearbox system vibration response is given in Figure 9.

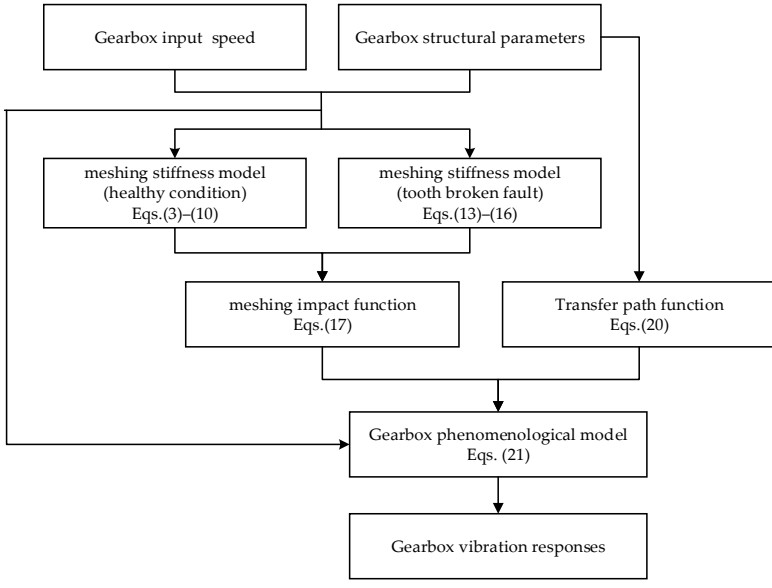

**Figure 9.** Flowchart showing the numerical analysis process.

Based on Equation (21), a simulated signal and its spectrum of a planetary gearbox under healthy conditions are shown in Figure 9. Three amplitude modulation envelopes of the planet carrier in a revolution period can be seen when the planet gear passes directly below the sensor in sequence. Moreover, Figure 10 shows a vibration signal and its spectrum of a conventional phenomenological model constructed by cosine functions [15].

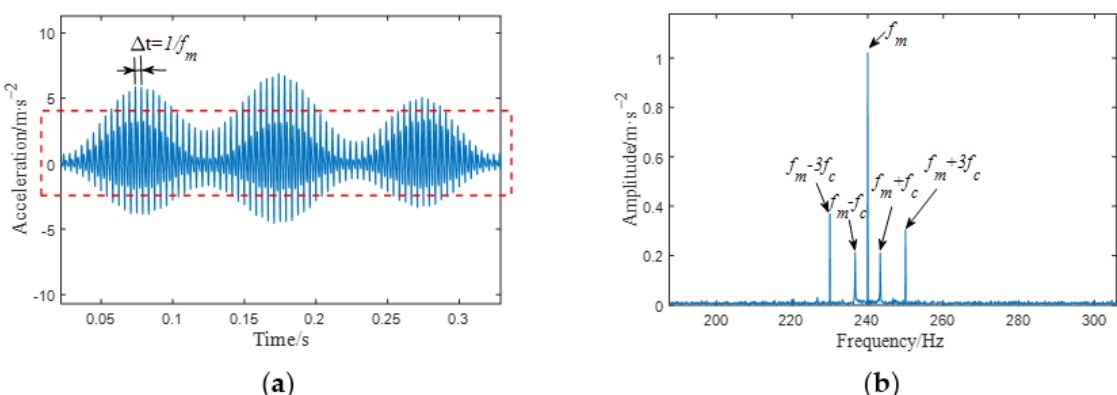

**Figure 10.** A simulated vibration signal of the proposed phenomenological models considering meshing impacts: (**a**) waveform; (**b**) frequency spectrum.

Comparing Figure 10 with Figure 11, it can be seen that the meshing impact force leads to apparent vibration impacts in the time-domain waveform in Figure 10a, and each shock interval is the reciprocal of the gear meshing frequency $1/f_m$. In contrast, the three envelopes shown in Figure 10a cannot reflect the impact characteristics caused by the meshing shock. The meshing frequency is observed in Figures 10b and 11b, and additional frequency components appear at $f_m - 3f_c$, $f_m - f_c$, $f_m + f_c$, $f_m + 3f_c$. However, its amplitudes and sidebands are significantly different; for symmetric sidebands with $nf_c$ intervals spaced around the meshing frequency in Figure 10b, the meshing frequency and its sidebands are significantly lower in amplitude, and, conversely, the sideband $f_m + f_c$ is almost nonexistent in Figure 11b. The comparison of the simulation results shows that the phenomenological model proposed in this paper can more accurately characterize the impact characteristics of the actual signal.

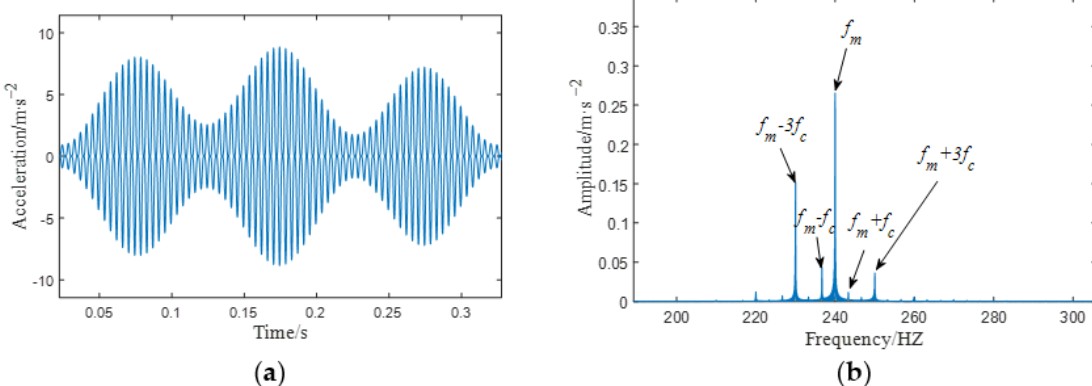

**Figure 11.** A simulated vibration signal of conventional phenomenological models constructed by cosine functions: (**a**) waveform; (**b**) frequency spectrum.

### 3.2. Phenomenological Model of Planetary Gearbox under Fault Condition

When a broken fault occurs on the sun gear, it can lead to a remarkable decrease in the mesh stiffness, increase the vibration amplitude, and intensify the impact effects. Replacing $k_{total}$ in Equation (12) with $k_{total,fault}$, the meshing shock $R(t)$ under fault condition is expressed as follows:

$$R(t) = \frac{F}{\sqrt{k_{total,fault}(t)}} \left( \frac{1}{m_{e1}} + \frac{1}{m_{e2}} \right) e^{-\xi wt} \sin(\omega t). \tag{22}$$

The vibration of the sun-planet gear pair can be expressed as follows [25]:

$$v_{spi}(t) = \sum_{k=-K}^{K} \sum_{\substack{q=-Q \\ q \neq 0}}^{Q} \overline{c_k} V_q^{sp} e^{j\varphi_q} e^{j2\pi(qf_m + kf_c)t}, \tag{23}$$

where $V_q^{sp}$ is the Fourier coefficients of $v_q^{sp}$, $v_q^{sp}$ is the amplitude of the $q^{th}$ harmonic of $v^{spi}(t)|_{n=1,...,N}$, and $\overline{c_k}$ is a coefficients of Fourier series.

According to the meshing stiffness under fault conditions, other parameters are kept constant, and the phenomenological model with broken sun gear fault vibration signal can be represented as follows:

$$x(t) = \sum_{i=1}^{N} \left( v_{rpi}(t) + v_{spi}(t) \right) \cdot \Phi_f \cdot W_{pi}(t), \tag{24}$$

where $\Phi_f$ is the meshing impact effect under broken fault condition.

Based on Equation (24), Figure 12 depicts the simulated vibration signal and its spectrum caused by the faulty sun gear. Similarly, three equally spaced envelopes of the planet carrier exist in a revolution period. However, the impact due to the tooth fault will emerge three times after the planet gear rotates for one cycle. The obvious impacts with an interval of $1/f_{rs}$ caused by the sun gear fault emerge in the time-domain waveform, as shown in Figure 12a. Figure 12b shows that the same meshing frequency $f_m$, $2f_m$, $3f_m$ also exist. Moreover, the vibration signal appears as sidebands with $f_{rs}$ interval under the fault condition.

To further analyze the variation of vibration characteristics under different degrees of broken tooth, Figure 13 shows the frequency spectrum of the gearbox vibration signal with different fault sizes. In Figure 13, the fault frequency of sun gear $f_{rs}$ can be observed clearly, and there is a symmetric sideband spaced by the rotational frequency of planet carrier $f_c$, which indicates the necessity of considering the time-varying transfer path modulation and shows that the proposed phenomenological model can effectively simulate the sun gear broken fault. The amplification near the peak of fault frequency $f_{rs}$ shows that the amplitude of the frequencies related to the fault increase gradually with the increase in the fault size, which verifies the validity of the signal model from the perspective of frequency.

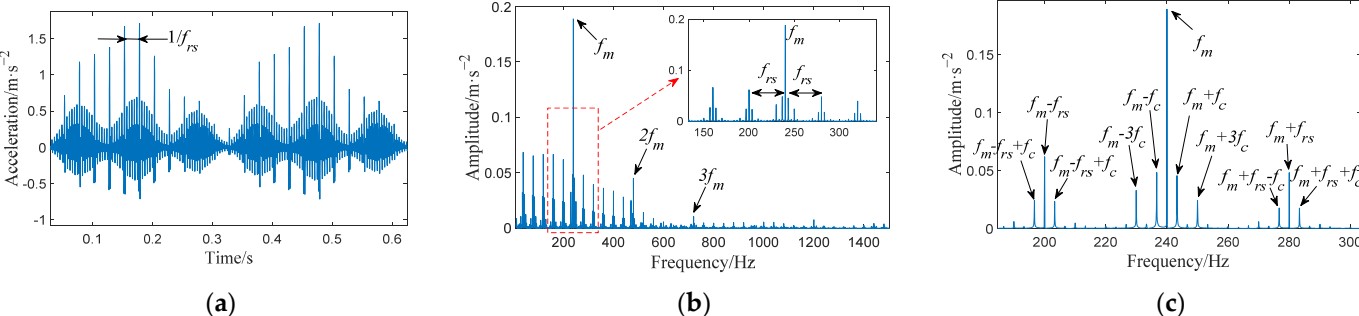

**Figure 12.** Simulated signal of the planetary gearbox with a broken tooth on the sun gear: (**a**) waveform; (**b**) frequency spectrum; (**c**) spectrum near the meshing frequency.

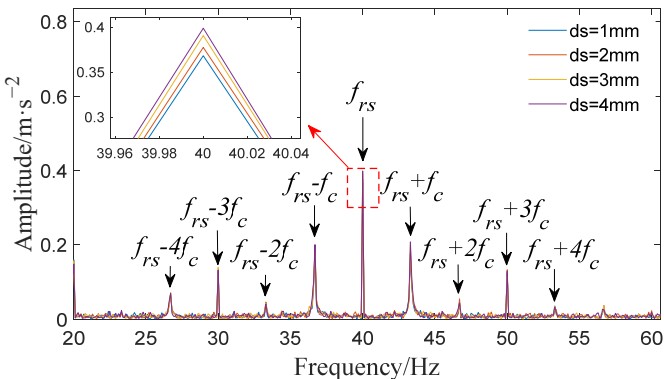

**Figure 13.** Frequency domain curves of sun gear with different broken sizes.

## 4. Experiment and Analysis of Experimental Results

### 4.1. Introduction of the Test Rig

In order to compare the simulation results with those obtained experimentally, the simulation parameters are identical to the test rig parameters, which are listed in Table 2. The experimental platform shown in Figure 14a is used in this paper to carry out our experimental verification. The testbed is mainly composed of four parts: the driving motor, the speed sensor, the planetary gearbox, and the brake. The motor is used as the input to drive the entire system. Three planet gears are equally spaced in the gearbox and share the load from the brake. All parts are connected by the elastic couplings, which effectively reduces the influence of minor manufacturing and installation errors. The faulty sun gear is shown in Figure 14c. During the experiment, the input speed of the motor is 1000 rpm, the sampling frequency is 12,800 Hz. The characteristic frequencies of the planetary gearbox are listed in Table 3.

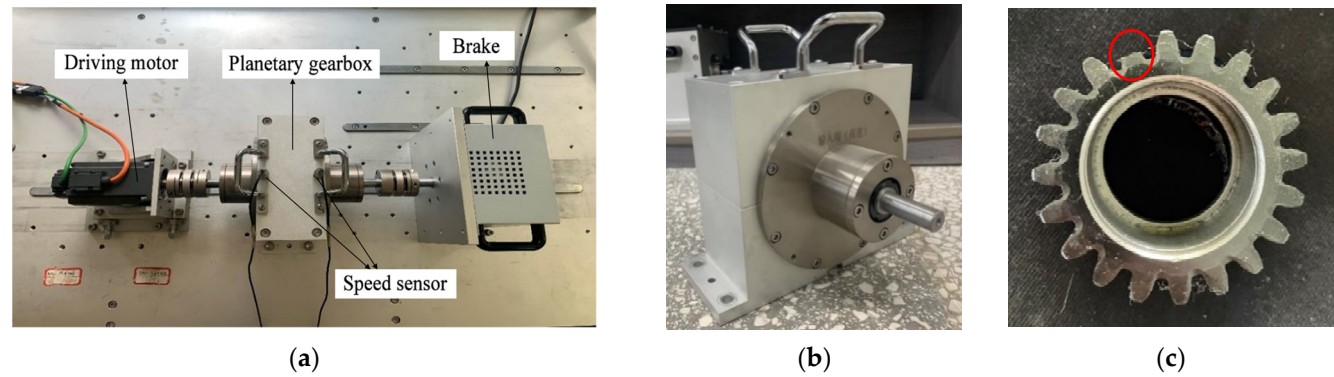

**Figure 14.** Experimental platform and specimens: (**a**) planetary gearbox test rig; (**b**) the planetary gearbox; (**c**) the sun gear with 3 mm fault.

**Table 2.** Assembled parameters of the planetary gearbox.

| Item | Sun Gear | Plant Gear | Ring Gear |
|---|---|---|---|
| Tooth number | 18 | 27 | 72 |
| Module(mm) | | 2 | |
| Width of teeth (mm) | | 20 | |
| Addendum coefficient | | 1 | |
| Pressure angle (°) | | 20 | |
| Young's modulus (Pa) | | $2.06 \times 10^7$ | |
| Poisson's ratio | | 0.3 | |

**Table 3.** Values of characteristic frequencies of planetary gearbox.

| Item | Symbolization | Value |
|---|---|---|
| Meshing frequency | $f_m$ | 240 Hz |
| Rotational frequency of planet carrier | $f_c$ | 3.34 Hz |
| Rotational frequency of sun gear | $f_s$ | 16.67 Hz |
| Fault frequency of sun gear | $f_{rs}$ | 40 Hz |

*4.2. Comparison Analysis between Simulated and Measured Vibration Signals under Healthy Condition*

Figure 15 illustrates the waveform and frequency spectrum of the simulated and measured vibration signals of the planetary gearbox when all the gears are perfect. By using Equation (25), the waveform error between the simulated signal and the measured signal is 1.1361%. Comparing Figure 15c,d, the gear meshing frequencies $f_m$, $2f_m$, and $3f_m$ can be observed whether in the simulated signal or the measured signal. Moreover, major sideband components are annotated in the frequency spectrum. These sidebands are located on both sides of the meshing frequency, which are consistent with the results of the theoretical analysis in Section 3.1.

$$e = (y(t) - \widetilde{y}(t))/y(t), \tag{25}$$

where $e$ is the relative error of waveform distortion, $y(t)$ is the vibration signal obtained from simulation, and $\widetilde{y}(t)$ is the vibration signal obtained from actual measurement. The error between the theoretical, simulated, and measured values is calculated and listed in Table 4. As shown in Table 4, the meshing frequency obtained from the proposed model is the same as the theoretical value, and the error with the measured meshing frequency is only 0.025%. This paper refers to the model verification methods in the literature [24,25], which use the time–frequency domain comparison method to verify the proposed model. Through the simulation and experimental signal analysis, we believe that the proposed phenomenological model is correct.

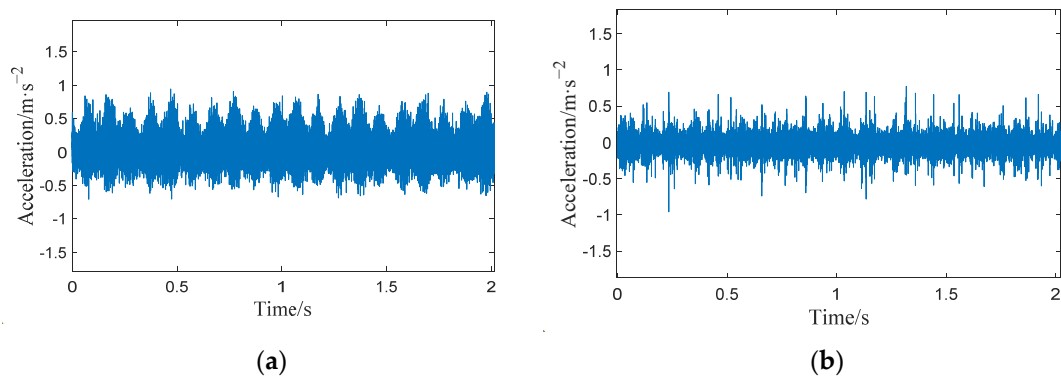

(**a**)　　　　　　　　　　　　　　　　(**b**)

**Figure 15.** *Cont*.

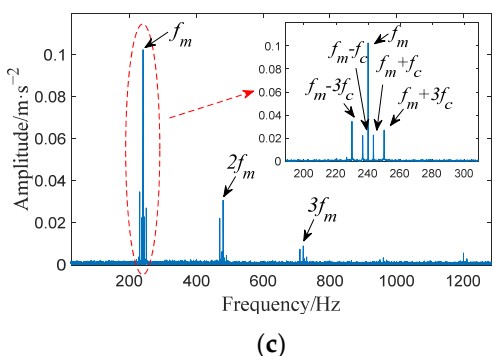

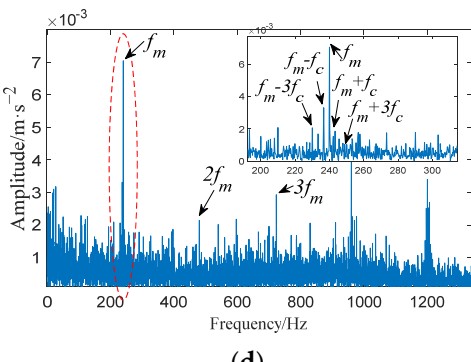

**Figure 15.** Comparison of simulated and measured signal under healthy condition: (**a**) simulated signal; (**b**) measured signal; (**c**) frequency spectrum of simulated signal; (**d**) frequency spectrum of measured signal.

**Table 4.** Comparative analysis of the simulation and measured results for meshing frequency.

| Theoretical Value | Simulated Value | Measured Value | *e* |
|---|---|---|---|
| 240 Hz | 240 Hz | 239.94 Hz | 0.025% |

### 4.3. Comparison Analysis between Simulated and Measured Vibration Signals under Fault Condition

Figure 16 draws the waveform and frequency spectrum of the simulated and measured vibration signal of the planetary gearbox with one tooth broken on the sun gear. Similarly, the waveform error between the simulated signal and the measured signal is 1.1361%. The gear meshing frequencies $f_m$, $2f_m$, and $3f_m$ can be observed in Figure 16c,d. However, it can be seen that due to the tooth fault, the sidebands are abundant and the spectral structure turns out to be more complex than that of the healthy condition. These sidebands appear at the following locations: $f_m - f_{rs} - f_c$, $f_m - f_{rs}$, $f_m + f_{rs}$, $f_m - f_{rs} + f_c$, which can help diagnose the fault through the spectrum. These results are consistent with the results of the theoretical analysis in Section 3.2. It should be mentioned that there exist some difference between simulation signal and experimental signal pertaining to the aspect of amplitude; this is inevitable because the given parameters of the simulation signal are hard to perfectly match the actual condition of the experimental planetary gearbox. However, the amplitude difference does not affect the verification of the proposed model, which focuses on the frequency features. In general, the simulated signal's characteristics are consistent with the measured signal, which proves the correctness of the proposed model.

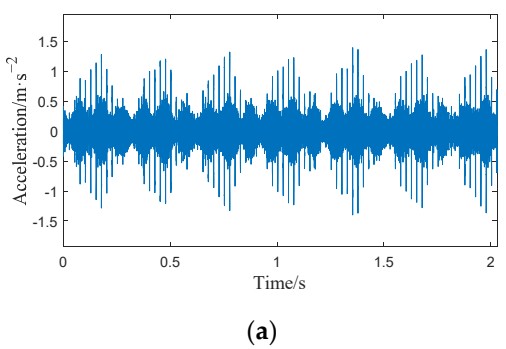

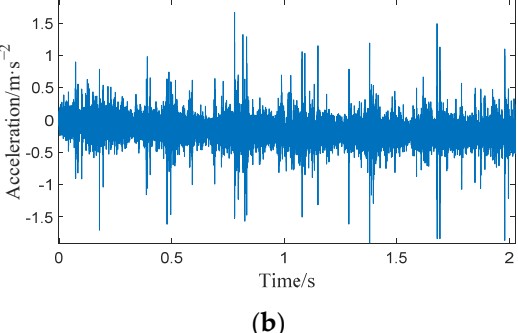

**Figure 16.** *Cont.*

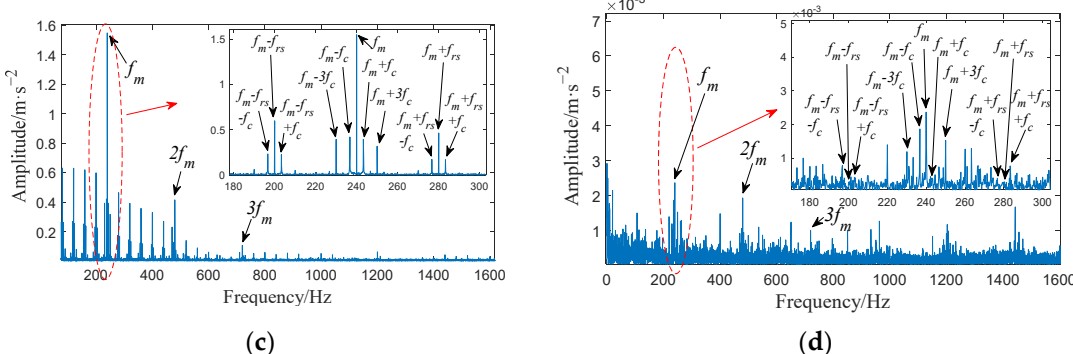

**Figure 16.** Sun gear tooth with 3 mm broken tooth failure: (**a**) simulated signal; (**b**) measured signal; (**c**) frequency spectrum of simulated signal; (**d**) frequency spectrum of measured signal.

### 4.4. Comparative of the Model Descriptive Capability

To verify the correctness of the above analysis, a comparative study between the traditional models and the proposed model will be given. Referring to ref. [24], the simulation result is shown in Figure 17. By using Equation (25), the waveform error of literature [24] is 2.5189%, and that of this paper is 1.1361%, which is reduced by 1.38%. Comparing Figure 17c,d, the meshing frequencies $f_m$ and $2f_m$ can be observed clearly, and some sidebands occur at the frequency locations of $f_m \pm f_c$, $f_m \pm 3f_c$. However, the frequency amplitude of the phenomenological model constructed in this paper is 92.5% higher than that of the literature [24]. To summarize the foregoing, the proposed model is preferred for its ability to describe the actual vibration of planetary gearbox.

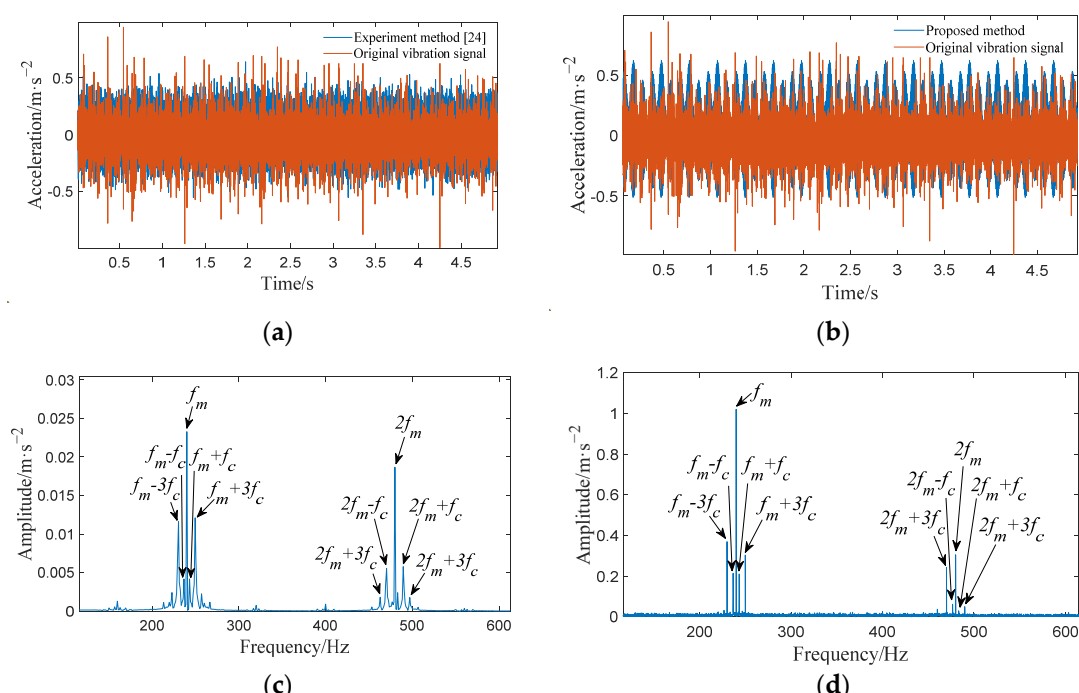

**Figure 17.** Comparison of the experimental method [24] and proposed method under healthy conditions: (**a**) vibration signal of experimental method [24]; (**b**) vibration signal of proposed method; (**c**) frequency spectrum of experimental method [24]; (**d**) frequency spectrum of proposed method.

As shown in Figure 18, a comparison was made between the simulated signal and the measured signals at three different rotational speeds when the sun gear had a broken tooth. The resulting waveform errors were found to be 0.61%, 1.14%, and 0.35%, respectively, all within the range of 2%. Moreover, the variance of the three error values was 0.0013, indicat-

ing that the model exhibits relatively stable error behavior during speed variations. This observation suggests that the model demonstrates robustness across different rotational speeds, yielding dependable results.

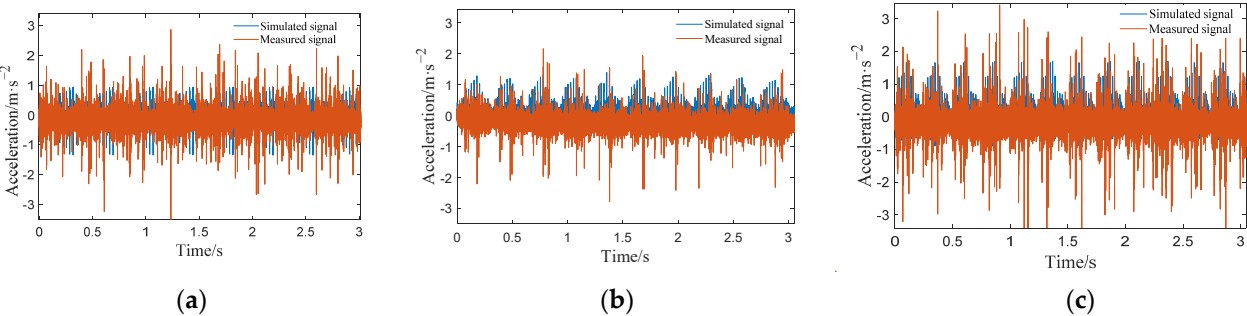

**Figure 18.** Comparison of measured and simulated signals at different rotational speeds under fault condition: (**a**) *v* = 800 rpm; (**b**) *v* = 1000 rpm; (**c**) *v* = 1200 rpm.

## 5. Discussion and Conclusions

### 5.1. Discussion

In this paper, the effects of the degree of broken teeth on the time-varying meshing stiffness and vibration of gears are discussed. Through simulation and experimental analysis, we evaluate the accuracy of the proposed model. Under the broken fault condition, the relative error of the waveform between the simulated signal and the actual signal at three speeds are calculated, respectively. The results show that the proposed model has good robustness with respect to the speed. Moreover, in the actual operation of the gearbox, there are many nonlinear factors, such as shaft misalignment, tooth backlash, friction force and temperature, etc. Owing to the shaft misalignment and tooth backlash, the internal dynamic excitations of gear pair will change compared with the ideal meshing condition, which will lead to variation in the system vibration. The friction force will cause time-varying meshing stiffness to slightly decrease. In addition, the temperature will cause the gear contact stress to increase, which will increase the wear between the tooth surfaces. We will also consider more influencing factors in order to improve our research in the future.

### 5.2. Conclusions

This paper proposes a phenomenological model considering the base deflection of the gear fillet and the influence of the flexibility between the root circle and the base circle, the transmission path, and the meshing shock for planetary gears. The meshing stiffness and vibration response of a gear with different fault sizes were investigated. The conclusions of this paper are as follows:

(1) When calculating the mesh stiffness via the potential energy method, the flexibility between the root circle and the base circle should be considered.

(2) The movement of the falling edge with the stiffness of the faulty gear pair caused by the fault shows a tendency to advance as the fault size increases. When the fault size is large ($\geq 1/2$ of tooth height), the stiffness of the faulty gear pair may be 0, leading to the unstable state of the gear system.

(3) Meshing impact is an important vibration excitation in the planetary gearbox. Compared with the traditional phenomenological model constructed by a series of cosine functions, the phenomenological model established in this paper considers the influence of the meshing impact and obtains a simulation signal that is more in line with the time–frequency domain characteristics of the actual signal.

(4) Under healthy conditions, the frequency components at the meshing point of the gearbox are the meshing frequency $f_m$ and its frequency doubling; the amplitude shows a gradual decreasing trend; and the sidebands appear at $f_m \pm f_c$, $f_m \pm 3f_c$. Under the sun gear broken tooth fault, the same frequency component and sidebands

appear at the meshing point as in the healthy case. In addition, there are also sidebands with the sun gear fault frequency $f_{rs}$ as the interval near the meshing frequency under the fault condition, and a symmetrical sideband with $f_c$ as the interval appears on both sides of $f_{rs}$. Analyzing the vibration signal characteristics of the planetary gearbox under normal conditions and with sun gear broken tooth faults is helpful in the local fault diagnosis of the planetary gearbox.

**Author Contributions:** This paper was completed by the authors in cooperation. M.Z. carried out theoretical research, data analysis, experiment analysis, and paper writing; J.M., X.X. and R.L. provided constructive suggestions; and J.M. revised the paper. All authors have read and agreed to the published version of the manuscript.

**Funding:** This research was supported by the National Natural Science Foundation of China (No.62163020, No.62173168) and Kunming University of Science and Technology 'Double First-Class' Science and Technology Project (No. 202202AG050002).

**Data Availability Statement:** Not applicable.

**Conflicts of Interest:** The authors declare no conflict of interest.

### Nomenclature

| | | | |
|---|---|---|---|
| $F$ | The meshing force | $v$ | Poisson's ratio |
| $R_b$ | Base circle radius | $E$ | Young's modulus |
| $R_f$ | Root circle radius | $L$ | Width of gear |
| $k_a$ | Axial compression stiffness | $G$ | Shear modulus |
| $k_b$ | Bending stiffness | $A_x$ | Cross-sectional area of the section in X direction |
| $k_s$ | Shear stiffness | $I_x$ | Area moment of inertia |
| $k_h$ | Hertz contact stiffness | $ds$ | Size of the broken tooth |
| $k_f$ | Matrix stiffness | $v_{rpi}(t)$ | Vibration generated at the meshing point between the ring and the planet i |
| $c, s, r, p$ | Planet carrier, sun gear, ring gear, and planet gear | $v_{spi}(t)$ | Vibration generated at the meshing point between the sun and the planet i |
| $f_{rs}$ | Fault frequency of sun gear | $v_q^{sp}$ | Vibration amplitude of the $q^{th}$ harmonic of $v^{spi}(t)\|_{n=1,\dots,N}$ |
| $f_c, f_s, f_r, f_p$ | Rotational frequencies of the planet carrier, sun gear, ring gear, and planet gear | PG | Planetary gearbox |
| $F_a$ | The meshing force is divided in the X-direction | LPDM | Lumped-parameter dynamic model |
| $F_b$ | The meshing force is divided in the Y-direction | | |

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
