# Peer review of "Analysis of Vibration Characteristics of Planetary Gearbox with Broken Sun Gear Based on Phenomenological Model"

_applsci, doi:10.3390/app13169413_

Round 1
Reviewer 1 Report
Explain whether backlash is considered in the model and its impact on the vibration spectrum. Address how the phenomenological model handles gear backlash, which is essential for accurately predicting vibration behavior in gear systems.
Discuss the effects of gear misalignment on the vibration behavior of the planetary gears. Misalignments can significantly impact gear kinematics and introduce additional vibration sources, so their consideration in the model is crucial for real-world applicability.
Elaborate on treating geometric errors and manufacturing tolerances in the gear system. These imperfections can affect gear meshing and introduce vibration sources, so their inclusion in the model may improve accuracy.
How are the dynamic forces during gear meshing, such as the impact forces, incorporated into the kinematic model? Properly accounting for these dynamic forces is essential for accurately predicting vibration spectra.
What is the influence of lubrication on gear kinematics and vibration? Lubrication can affect mesh stiffness, damping, and frictional forces, which may influence the vibration behavior of the gears.
If temperature variations are significant in the gearbox operation, how do these changes affect gear kinematics and vibration characteristics? Temperature-induced dimensional changes can impact gear tooth engagement and mesh stiffness.
The paper should offer a strong rationale for considering specific components in the phenomenological model, such as the base deflection of the gear fillet and the flexibility between the root circle and the base circle.
Explain why these elements are critical for accurately representing time-varying mesh stiffness and broken tooth faults.
What are the proposed model's practical implications, and how can it be applied in real-world scenarios for local fault diagnosis of planetary gearboxes? The authors should highlight potential industrial applications and address any model implementation challenges.
A sensitivity analysis should be performed to assess the model's robustness concerning variations in the input parameters. This analysis will reveal which parameters have the most significant impact on the model's results and conclusions.
State the limitations of the proposed model concerning its applicability to different types of planetary gearboxes (e.g., varying gear ratios, sizes, and manufacturing tolerances). Discuss the generalizability of the model to diverse gearbox configurations.
Minor editing of English language required
Author Response
Dear expert,
We gratefully appreciate you for your time spend making positive and constructive comments, These comments are all valuable and helpful for revising and improving our manuscript entitled “Analysis of vibration characteristics of planetary gearbox with broken sun gear based on phenomenological model”(ID: applsci-2534715), as well as the important guiding significance to our researches.
We have studied comments carefully and have made the corrections which we hope would the criteria. Revised portion are highlighted in red in the revised manuscript. Hope that the changes we’ve made resolve all your concerns about the article. The details are in the attachment.

Reviewer 2 Report
Authors investigated the base deflection of the gear fillet and the influence of the flexibility between the root circle and base circle, transmission path, and meshing shock are considered to establish a more realistic phenomenological model for planetary gears. The paper is professionally written and interesting to read, however I see the following major issues that should be resolved before publishing this paper:
The authors should enlarge the introduction section by adding new references specially published in 2023.
The authors should validate the tooth and mesh stiffness results by comparing some reliable results in literature.
The authors should present all gear parameters.
The abstract and the conclusion seem inconsistent.
A flowchart for the proposed model should be added.
Author Response

(The authors gave the same response as above.)

Reviewer 3 Report
Dear Authors,
The Authors undertook a very difficult task:
a) analytical determination of the tooth engagement stiffness without cracking and with fatigue cracking (up to breakage),
b) developing a phenomenological model of tooth vibrations,
c) simulation of vibrations of undamaged and damaged teeth of sun gear
d) elaboration of simulation results (modulation, determination of the frequency spectrum,
e) conducting an experiment
f) comparison of the results of the numerical simulation with the results of the experiment.
Although the problem of using the phenomenological model of tooth vibrations of gear wheels in diagnostics has already been presented in several works (not only in those cited by the authors [19] and [20], it is good that the Authors tried to conduct their own research. However, it is not possible to check the stiffness of the meshing, because the Authors did not provide the values of the coefficients characterizing the degree of tooth damage, they referred to literature [16], where the values of these coefficients are actually given. However, the values given in [16] (Table 1) apply to other sizes of gears, because the stiffness of healthy teeth differ as much as 100 times. Since the rest of the manuscript is done carefully, Authors should disclose the coefficient values and parameters (modulus, wheel widths, etc. as in [16]). According to [16] the coefficients can be approached by polynomial functions as in paper of P. Saist and P. Velex cited below.
Additional Note: Authors should cite the very interesting paper of Ma J., Liu T., et.al. , where more accessible methods for stiffness determination and vibration simulation are presented.
Sincerely, Reviewer
1) Ma, J.; Liu, T.; Zha, C.; Song, L. Simulation Research on the Time-Varying Meshing Stiffness and Vibration Response of Micro-Cracks in Gears under Variable Tooth Shape Parameters. Appl. Sci. 2019, 9, 1512; doi:10.3390/app9071512
2) P. Sainsot, P. Velex, O. Duverger. Contribution of Gear Body to Tooth Deflections—A New Bidimensional Analytical Formula. J. Mech. Des. Jul 2004, 126(4): 748-752
Author Response

(The authors gave the same response as above.)

Reviewer 4 Report
Dear Authors,
First of all accept my sincere congratulations for this great research and very elegant model. I consider this paper a very valorous contribution in the field of gear transmissions. However, I have done some observations and I listed some suggestion in the attachment.

Author Response

(The authors gave the same response as above.)

Reviewer 5 Report
The paper is well written and it needs minor revision. My comments are in below:
(1) The paper needs nomenclature.
(2) In figure (8a) the author calculate 1/fm using the acceleration. Why the author did not use the displacement instead. Is the fm in the frequency spectrum figure (8b) is the fundamental or the dominant frequency. Please explain.
(3) Did the author use the acceleration varying with time in the tool of frequency spectrum analysis of Fast Fourier Transform?
(4) The experiment results in figure (13d) of frequency spectrum need filtration.
(5) How the author measured the signal of vibration experimentally?
(6) In table (3) how the author measure the meshing frequency numerically and experimentally? In the simulation analysis did the author use ANSYS program ?
(7) The conclusion section needs more attention.
Author Response

(The authors gave the same response as above.)

Round 2
Reviewer 1 Report
The authors have addressed all my comments positively. The paper can be accepted.
Author Response
Dear reviewer,
Thank you very much for reviewing the manuscript. Thanks for your patience and helpful suggestions.
Thank you again for your positive comments on our manuscript.
Reviewer 2 Report
The authors have addressed all my comments for this paper and answered the technical questions I have for this method. The paper has been significantly improved after revising. The revised manuscript is ready for publication.
Author Response

(The authors gave the same response as above.)

Reviewer 3 Report
Dear Authors,
After corrections, your manuscript appeared to be correctly written. However, after comparing the geometrical parameters of all the gears adopted by the authors of the manuscript with the data given in paper [20] on which the authors were based, it can be concluded that only the m=2 modules are equal. However, the numbers of teeth differ significantly, as the authors assumed z1=18 and z2 = 27 <42, while in paper [20] z1=25 and z2=53 >42. The number of teeth equal to z=42 is to be according to [20], i.e. the limit number of two different computational models for the analysis of tooth cracking at the base (I emphasize - fatigue cracking). It can be mentioned here that the attempt to prove is weak.
In the theory of gears (e.g. J.E. Shigley, C.R. Mischke -Mechanical Engineering Design Mc Graw Hill) a relatively simple geometric model and the Lewis equation derived on the basis of this model are known. From the model and the equation, the Lewis form factor can be determined by a graphical layout of the gear tooth or by computer calculation, respectively. The analytically determined Lewis form factor is used in the ANSI/AGMA 2001-C95 and ISO 6336-3 standards. Of course, FEM was confirmed earlier, i.e. at the point of contact of the straight line (inclined to the vertical at an angle of 30 degrees) with the root fillet, the maximum tangential stresses (to the root fillet) actually occur, i.e. at this point (on the side of the tooth with more load) may begin fatigue cracking. Also these 2 symmetrically located points determine the thickness of the tooth root in the design cross-section (measured along the chord).
The Authors of the manuscript deal with fracture of the tooth above the pitch circle (at a distance of 1 mm and 2 mm from the apex), i.e. impact fracture. Sometimes fatigue cracking of the tooth in the pitch circle zone may also occur, when the source of crack propagation are pits formed in this zone, i.e. at a distance of as much as 2.5 mm from the pitch circle to the circle of feet (at the modulus m=2, the number of teeth z1=18, shift coefficient x1=0 and alpha=20 degrees). In paper [20], the case of tooth cracking outside the tooth foot is not considered (even nothing mentioned), so in the reviewer's opinion there is no reason to include models from paper [20 ] in a peer-reviewed manuscript.
Hence the conclusion to save the interesting (after corrections) manuscript: According to the reviewer, the authors should replace the modified method from paper [20] with the classical analytical method without considering the zone between the foot and base circles.
The reviewer would additionally ask for: (1) the Authors' response to the allegation due to the application of a modified method to their cracking case, (2) the answer from where the Authors obtained the planet gear wheel (as in [20]) with the number of teeth greater than 42 to plot the diagram shown in Figure 3b and (3) please specify the angular velocity of the sun gear, because the horizontal axis (Figure 6a,b,c,d) in the mesh stiffness diagrams is expressed in seconds and not in degrees of rotation angles ?
Sincerely, Reviewer
Author Response
Dear reviewer,
We gratefully appreciate you for your time spend making positive and constructive comments. These comments are all valuable and helpful for revising and improving our manuscript, as well as the important guiding significance to our researches.
We have studied comments carefully and have made the corrections which we hope would the criteria. Revised portion are highlighted in blue in the revised manuscript. Hope that the changes we’ve made resolve all your concerns about the article. The details are in the attachment. Please see the attachment.
Kind regards,
Mengting Zou

Round 3
Reviewer 3 Report
Dear Authors v.2
Thank you very much for your comprehensive responses to my comments in the review of your manuscript. I was not satisfied with only two answers regarding the limit number of teeth z=42 and referring to the comparison of mesh stiffness graphs obtained by analytical methods and FEM. . I will justify my dissatisfaction. Namely, in paper [19] attached by the Authors in the cover letter paper, it was explained that z=42 was determined from the condition of equality of the pitch and base diameters for the profile shift coefficient x=0. Today, in practice, x=0 is rarely assumed, most often due to the need to select the gears to the imposed axis distances (as, for example, when constructing the size range of gears). Then gears with z=42 teeth can have pitch diameters smaller or larger than the base diameter for different values of x.
Three examples are presented below.
Data:
,
,
,
,
,
(first example),
, where
, accepted
(second example) (third example) accepted
.
,
First example
:
.
Third example
:
![]()
Hence the conclusion, you can adopt a model for determining the mesh stiffness depending on which of the diameters is greater P>B or B>P). But you can't be guided by the number of teeth z=42 (see paper Xihui Liang, Ming J. Zuo, Mayank Pandey. Analytically evaluating the influence of crack on the mesh stiffness of a planetary gear set. Mechanism and Machine Theory 76 (2014) 20–38).
The second issue concerns the Authors' reference to a paper in which mesh stiffness graphs obtained by analytical methods and FEM were compared without presenting the FEM model. Because correct models of tooth contact and mesh stiffness determination (such as in paper An improved analytical method for mesh stiffness calculation of spur gears with tip relief. Hui Ma, Jin Zeng, Ranjiao Feng, Xu Pang, Bangchun Wen. Mechanism and Machine Theory 98 (2016) 64–80) are not often found in the literature.
In conclusion, I think that the Authors should remove the limit number of teeth z=42, staying with the model matched to the number of teeth 18 and 27 (outer gears) and 72 (inner gear). Although the mesh stiffness of the inner and outer gear pairs is mentioned by the authors a little.
Sincerely, Reviewer

Author Response
Dear reviewer,
Thank you again for your careful reading and valuable guidance of our paper. We have studied comments carefully and have made the corrections which we hope would the criteria. Revised portion are highlighted in blue in the revised manuscript. Hope that the changes we’ve made resolve all your concerns about the article. Please see the attachment.
Kind regards,
Mengting Zou
